# Canine Mesenchymal-Stem-Cell-Derived Extracellular Vesicles Attenuate Atopic Dermatitis

**DOI:** 10.3390/ani13132215

**Published:** 2023-07-06

**Authors:** Byong Seung Cho, Sung-Bae Kim, Sokho Kim, Beomseok Rhee, Jungho Yoon, Jae Won Lee

**Affiliations:** 1ExoCoBio Exosome Institute (EEI), ExoCoBio Inc., Seoul 08594, Republic of Korea; ceo@exocobio.com; 2Korea Conformity Laboratories, Incheon 21999, Republic of Korea; suaa10@kcl.re.kr; 3Research Center, HLB bioStep Co., Ltd., Incheon 22014, Republic of Korea; skim@hlbbiostep.com (S.K.); beomseok@hlbbiostep.com (B.R.); 4Department of Veterinary Medical Imaging, College of Veterinary Medicine, Chungnam National University, Daejeon 34134, Republic of Korea; 5Equine Clinic, Jeju Regional Headquarter, Korea Racing Authority, Jeju 63346, Republic of Korea; junghoy11@gmail.com

**Keywords:** canine atopic dermatitis, extracellular vesicles, mesenchymal stem cells, adipose tissue

## Abstract

**Simple Summary:**

Atopic dermatitis (AD) is a chronic inflammatory skin disease associated with systemic inflammation and immune modulation. Previous studies showed that extracellular vesicles from human adipose-tissue-derived mesenchymal stem cells reduced inflammatory cytokines and attenuated AD-like symptoms. In this study, we aimed to investigate the effects of canine ASC-exosomes on canine AD using a Biostir-induced AD mouse model. The study found that cASC-EVs improved AD-like dermatitis, decreased serum IgE, ear thickness, and inflammatory cytokines such as IL-4 and IFN-γ in a dose-dependent manner. The study also conducted in vivo toxicity studies in ICR mice and found no systemic toxicity. miRNA analysis using next-generation sequencing showed anti-inflammatory miRNAs present in cASC-EVs, which suggests a promising cell-free therapy for treating canine AD.

**Abstract:**

Atopic dermatitis (AD) is a chronic inflammatory skin disease that is associated with systemic inflammation and immune modulation. Previously, we have shown that extracellular vesicles resulting from human adipose-tissue-derived mesenchymal stem cells (ASC-EVs) attenuated AD-like symptoms by reducing the levels of multiple inflammatory cytokines. Here, we aimed to investigate the improvement of canine AD upon using canine ASC-exosomes in a Biostir-induced AD mouse model. Additionally, we conducted in vivo toxicity studies to determine whether they targeted organs and their potential toxicity. Firstly, we isolated canine ASCs (cASCs) from the adipose tissue of a canine and characterized the cASCs-EVs. Interestingly, we found that cASC-EVs improved AD-like dermatitis and markedly decreased the levels of serum IgE, ear thickness, inflammatory cytokines, and chemokines such as IL-4 and IFN-γ in a dose-dependent manner. Moreover, there was no systemic toxicity in single- or repeat-dose toxicity studies using ICR mice. In addition, we analyzed miRNA arrays from cASC-EVs using next-generation sequencing (NGS) to investigate the role of miRNAs in improving inflammatory responses. Collectively, our results suggest that cASC-EVs effectively attenuate AD by transporting anti-inflammatory miRNAs to atopic lesions alongside no toxicological findings, resulting in a promising cell-free therapeutic option for treating canine AD.

## 1. Introduction

Atopic dermatitis (AD) is a chronic and relapsing skin disorder characterized by cutaneous inflammation and defects in the epidermal barrier function [1]. It is one of the most common skin disorders, estimated to be present in up to 1–3% of adults and 20% of children worldwide [2]. The pathophysiology of AD remains unclear, although epidermal barrier dysfunction due to immunological responses and genetic defects plays an important role in the deterioration and development of AD [1].

AD in dogs is an allergic skin disease with a prevalence of 10–15% and many similarities with human AD [3]. The cause of the onset of canine AD remains unclear, although it is usually presumed to be due to skin barriers or immunological changes resulting from various interactions, such as an imbalance in immune function, as in humans [3,4,5]. Currently, canine AD drugs have different clinical efficacy in different breeds, potentially because multiple gene abnormalities and altered immunological processes can be involved [3,6]. In addition, steroids should be used for moderate or high forms of AD, which have serious side effects; therefore, the development of safe and effective drugs is continuously being studied [7]. Thus, there is a trend to utilize stem cell treatments or various new technologies for the treatment of canine AD.

Mesenchymal stem cells (MSCs) are self-regenerative cells with the potential to differentiate into multiple cell types [8]. MSCs are present in various tissues, such as bone marrow, fat, umbilical cord, and kidney, and can differentiate into osteoblasts, adipose cells, and muscle cells. [9,10] Since MSCs were established, cell therapy using them has been reported in various disease models, including autoimmune diseases [11], myocarditis [12], and glomerulonephritis [13]. MSCs also have the immunomodulatory ability to regulate the inhibition of Th2 cells and increase regulatory T (Treg) cells [14]. In particular, MSCs do not have major histocompatibility complex (MHC) II, while co-stimulatory molecules, such as CD80 and CD86 play an important role in allogeneic antigen recognition [15]. Therefore, because the immunogenicity of MSCs is relatively low, the therapeutic effect of immunomodulatory action can be clinically expected [16].

Recent studies have shown that MSCs exert an immunosuppressive effect by producing and releasing extracellular vesicles (EVs) of various sizes, which consist of lipid bilayers rather than through cell-to-cell contact [17]. EVs are considered essential carriers of cellular communication molecules, which encapsulate a variety of genetic materials. MSC-derived EVs (MSC-EVs) contain regulatory molecules capable of modulating immune cell functions [18]. MSC-EVs have also been shown to have immunomodulatory abilities similar to those found in MSCs [18].

EVs are nanosized vesicles (approximately 30–200 nm in size) that play an important role in cell-to-cell communications [17]. Alix and TSG101, which are known to exist inside EVs, and CD63, CD9, and CD81, which are present on the surface of EVs, are well-known as specific markers [18]. In addition, it is known that there is a difference in expression levels depending on the cell of origin. Stem cell-derived EVs contain a large number of molecules related to the regeneration/healing, anti-inflammatory, and immunomodulatory abilities of stem cells; therefore, research is underway for the development of a next-generation non-cell therapy [19]. Specifically, MSC-EVs have been shown to have broad anti-inflammatory and regenerative effects in an array of inflammatory disease models, including atopic disease [19]. 

It is difficult to accurately identify the trends in the animal cell therapy market because global market analysis has not been conducted properly. However, there are several confirmed reports on the therapeutic effects of EVs from adipose-derived stem cells in horses and dogs with arthritis [20]. Therefore, there is potential for using EVs to treat diseases in animals. 

In the present study, we isolated canine adipose stem cell (cASC)-derived EVs and characterized them. We also investigated whether cASC-EVs improved AD-like dermatitis in an animal model and addressed the safety concern in systemic toxicity studies using ICR mice. In addition, we performed next-generation sequencing (NGS) to study the role of miRNAs in improving inflammatory responses.

## 2. Materials and Methods

### 2.1. Isolation and Cultivation of Canine ASCs (cASCs)

Canine adipose tissue was obtained from Knotus Co., Ltd. (Incheon, Republic of Korea). Ten individual beagles at one year of age were used. In brief, the adipose tissue was chopped, and an aliquot was enzymatically digested at 37 °C for 1 h with 1% type 2 collagenase (Sigma, St. Louis, MO, USA) in phosphate-buffered saline (PBS), using a shaking incubator. The digested adipose tissue was centrifuged at 1000 rpm for 5 min, and the pellet was resuspended and passed through a 70 µm mesh filter (Cell Strainer, Becton Dickinson, Franklin Lakes, NJ, USA) to remove the debris. Cells were plated in 100 mm culture dishes at mononuclear cells with Dulbecco’s Modified Eagle Medium: Nutrient Mixture F-12 (DMEM/F12) containing 10% fetal bovine serum (FBS) and antibiotics (100 U/mL penicillin G and 100 µg/mL streptomycin). After one day, the medium was changed to remove non-adherent cells, while the adhered cells were expanded for five days and subcultured [21].

### 2.2. Fluorescence-Activated Cell Sorting (FACS) Analysis

Phenotypical characterization of cASCs (passage No. 3) was performed using a flow cytometer (Agilent, St. Clara, CA, USA, NovoCyte 3000). ASCs were removed via trypsinization, placed on ice for 30 min, and treated with the following labeled stemness-associated antigen markers: CD29+, CD44+, CD90+, CD105+, CD4−, CD8−, CD14−, CD25−, CD45−, CD80−, CD184−, and MHCII−. Other antibodies with the indicated specifications were purchased separately: CD29+ (Thermo Fisher Scientific, MA1-19458), CD44+ (Thermo Fisher Scientific, 11-5440-42), CD90+ (Thermo Fisher Scientific, 12-5900-42), CD4 (Thermo Fisher Scientific, MA5-16989), CD8 (Thermo Fisher Scientific, 17-5080-42), CD14 (BD, 555397), CD25 (Thermo Fisher Scientific, 63-0250-42), CD45 (Thermo Fisher Scientific, 48-5450-42), CD80− (Thermo Fisher Scientific, 46-0801-82), and CD184− (Thermo Fisher Scientific, 12-9991-82) [22].

### 2.3. Reverse Transcription Polymerase Chain Reaction (RT-PCR) for Phenotypical Characterization of cASCs

Total RNA was extracted from fresh cells (cASCs and CMT-U27; canine mammary cancer cell lines for comparison) using the RNA extraction Hybrid-R kit (GeneAll, 305-101). RNA concentration was measured at an absorbance of 260 nm with a spectrophotometer (Thermo Fisher Scientific, Waltham, MA, USA), and cDNA was synthesized from total RNA using the PrimeScript RT reagent kit with gDNA Eraser (TOYOBO, FSQ-301), according to the manufacturer’s protocol. The expression of specific genes was quantified by RT-PCR, in accordance with the instructions of i-StarMAX II ™ DNA Polymerase (iNtRON Biotechnology, 25173). The primer sequence is as follows. SOX2, F: ACAGCATGTCCTACTCGCAG, R: GGACTTGACCACCGAGCC; Nanog, F: CCAGACCTGGAACAGCCAAT, R: ACAGTTGTGGAGCGGATTGT; Oct4, F: GACACCTCCCAGCCGGA, R: TGCTCCAGCTTCTCCTTGTC; GAPDH, F: GTTTGTGATGGGCGTGAACC, R: TTTGGCTAGAGGAGCCAAGC.

For multipotency marker analysis, target genes were amplified at 94 °C (2 min), 35 cycles of 94 °C (10 s), 60 °C (10 s), 72 °C (10 s), followed by 72 °C for 7 min. PCR products were separated on a 2% agarose gel by electrophoresis, stained with Red Safe (iNtRON Biotechnology, Seongnam, Republic of Korea) and visualized under UV light. Images were digitally captured using an iBright™ CL1500 Imaging System (Invitrogen™, Waltham, MA, USA).

### 2.4. Osteogenic Differentiation: Alizarin Red Staining

Osteogenesis differentiation medium (Osteogenic Differentiation SingleQuotsTM Supplements Kit; Lonza) was used according to the manufacturer’s instructions. ASCs were cultured for 21 days, the medium was changed every 3rd day, and differentiation was assessed using alizarin red staining. For this process, the cells were fixed with 4% formaldehyde solution for 30 min, followed by rinsing with PBS, and incubation with alizarin red solution in the dark for 30 min. Then, the cells were washed several times with PBS and visualized under a light microscope. Red staining indicates the deposition of calcium phosphate precipitates by osteoblasts [23]. 

### 2.5. Chondrogenic Differentiation: Alcian Blue Staining

Chondrogenic differentiation medium (Chondrogenic SingleQuot Kit; Lonza, Basel, Swizerland) was used according to the manufacturer’s instructions. ASCs were cultured for 28 days, the medium was changed every 3rd day, and differentiation was assessed using Alcian blue staining. Cells were fixed with 4% formaldehyde for 30 min and washed with PBS. Then, 1% Alcian blue, which was prepared in 0.1 N HCl, was added for 30 min incubation, and distilled water was added. Blue staining indicated chondrocyte synthesis of proteoglycans [23]. Additionally, the pellet culture in 15 mL conical tubes was observed for 35 days. The pellet was stained after paraffin sectioning.

### 2.6. Adipogenic Differentiation: Oil Red O Staining

An adipogenesis differentiation kit (Adipogenic Induction SingleQuot Kit, Lonza) was used according to the manufacturer’s instructions. ASCs were cultured for 35 days. Cells were cultured with cASCs in supplemented adipogenesis induction medium and cultured for 3 days (37 °C and 5% CO_2_) followed by 1–3 days of culture in supplemented adipogenic maintenance medium. These cycles were repeated three times. The cells were cultured for 7 days in a supplemented adipogenic maintenance medium. Differentiation was assessed by the presence of lipid droplets, which were visualized after staining with Oil Red O solution. Cells were fixed with 10% formal calcium fixative for 60 min, washed with PBS, and then with 70% ethanol. The addition of Oil Red O solution was followed by rinsing the cells with 70% ethanol, followed by tap water. Red staining indicates the presence of lipids.

### 2.7. Real-Time Quantitative Polymerase Chain Reaction (RT-qPCR) for Analysis of Differentiation

Total RNA was extracted from fresh cells using an RNA extraction Hybrid-R kit (GeneAll, 305-101). RNA concentration was quantified by measuring absorbance at 260 nm with a spectrophotometer (Thermo Fisher Scientific, Waltham, MA, USA), and cDNA was synthesized from total RNA using a PrimeScript RT reagent kit with a gDNA Eraser (TOYOBO, FSQ-301), according to the manufacturer’s protocol. Then, the expression of our chosen genes was quantified by RT-PCR, in accordance with the instructions of the qPCRBIO SyGreen Blue Mix Separate-Rox (PCR Biosystems, PB20.17-05). The primer sequence is as follows. BMP2, F: CGGGAACAGATGCAGGAACC, R: AAAGTCTGGTCACGGGGAAC; RUNX2, F: TGCTTCATTCGCCTCACAAAC, R: GACTCTGTTGGTCTCGGTGG; OPN, F: AGGGACAGCCATGCAAAAGA, R: TACTCTTGGGAGTGCTTGCG; SOX9, F: CTACATGAACCCCGCGCAGA, R: GTGTGTAGACAGGCTGTTCCC; Aggrecan, F: AGAAGCCCTTCACTTTCGCC, R: CTCTCCAGTCCTGTTCTCGG; FAS, F: CTGCACGTCTTATGCGGGTA, R: TGCTCTCCATCGCAGATTCC; SREBP-1, F: TGCACGACTGCCAGCAAA, R: CGCGGACGGGGATCTA; GAPDH, F: GTTTGTGATGGGCGTGAACC, R: TTTGGCTAGAGGAGCCAAGC.

Reactions were performed using a CFX96 Touch Real-Time PCR Detection System (Bio-Rad, Hercules, CA, USA) with the following process steps: 95 °C for 2 min, 40 cycles of 95 °C for 10 s, and 60 °C for 20 s, followed by melting curve analysis. The specificities of the PCR products were verified by melting curve analyses between 65 and 95 °C. At the end of each reaction, CT values were obtained by analyzing the fluorescence data. Gene expression was calculated using the 2^−∆∆Ct^ method, where the values from different samples were averaged and calibrated in relation to GAPDH CT values.

### 2.8. Isolation of cASC-EVs

cASC-Evs were isolated from cASC-conditioned media (CM) by tangential flow filtration (TFF)-based ExoSCRT™ technology, as previously described [24]. Briefly, CM was filtered through a 0.22 μm polyethersulfone membrane filter (Merck Millipore, Billerica, MA, USA) to remove non-exosomal particles, such as cells, cell debris, microvesicles, and apoptotic bodies. Then, the CM was concentrated by tangential flow filtration with a 500 kDa molecular weight cut-off filter membrane cartridge (Cytiva, Chicago, IL, USA), and buffer exchange was performed by diafiltration with DPBS. The amount of protein in isolated cASC-EVs was approximately 0.5% of the amount of protein in the CM. Isolated cASC-EVs were aliquoted into polypropylene disposable tubes and stored at −80 °C until use. Before use, frozen cASC-Exos were left at 4 °C until completely thawed and were not frozen again. Characterization and profile analysis of the cASC-EVs were conducted following the Minimal Information for Studies of Extracellular Vesicles 2018 (MISEV2018), recommended by the International Society for Extracellular Vesicles (ISEV) [17].

The cASC-EVs used for analysis and AD treatment in this study were derived from passage 3 cASC and isolated from the CM collected under the same culture conditions for two days.

### 2.9. Quantification of cASC-EVs

To determine size distribution and particle concentration, cASC-EVs were diluted with DPBS and analyzed by nanoparticle tracking analysis (NTA) using a NanoSight NS300 (Malvern Panalytical, Amesbury, UK) equipped with a 642 nm laser. Then, the cASC-EVs were diluted with DPBS to between 20 and 80 particles per frame and scattered and illuminated by the laser beam, while their movements under Brownian motion were captured for 20 s each, at a camera level of 16. The subsequent videos were analyzed by NTA 3.2 software using constant settings. To provide a representative result, at least five videos were captured, and more than 2000 validated tracks were analyzed for each individual sample. The NTA instrument was regularly checked with 100 nm standard beads (Thermo Fisher Scientific). To provide a representative size distribution of the EVs, the size distribution profiles from each video replicate were averaged.

Protein quantification of cASC-EVs determined using the Micro BCA protein assay kit (Thermo Fisher Scientific, Waltham, MA, USA) according to the manufacturer’s protocol.

### 2.10. Bead-Based Flow Cytometric Analysis of Exosomal Surface Markers

The isolated cASC-EVs were captured and labeled with Dynabeads, according to the manufacturer’s instructions. Briefly, 2 μg of cASC-EVs were incubated overnight at 4 °C with capture beads. The captured EVs were labeled with a mixture of APC-conjugated anti-CD81 antibodies for 1 h at room temperature. The bead populations and APC intensities were analyzed using a NovoCyte 2000 R Flow Cytometer (ACEA Biosciences, San Diego, CA, USA), and the data were analyzed using NovoExpress software (ACEA Biosciences). The background was corrected with the median intensity of the anti-IgG-APC signals. Assays were performed in triplicate for three independent samples.

### 2.11. In Vivo Efficacy Study

#### 2.11.1. Animals and Study Design

Animal care and experimental procedures were approved by the Institutional Animal Care and Use Committee of the Korea Conformity Laboratories (IACUC number: IA20-02200). Male 6-week-old NC/Nga mice were obtained from Central Lab Animal, Inc. (Seoul, Republic of Korea). The mice were divided into six groups (*n* = 8 per group) as follows: the normal group (Ctrl group), Df-induced with no treatment group (AD group), Df-induced with EV treatment groups (1.00 × 10^9^, 3.33 × 10^9^, and 1.00 × 10^10^ particles/mL), and Df-induced with prednisolone treatment group (10 mg/kg). To induce AD-like skin lesions, 100 mg Dermatophagoides farinae ointment (Df; Biostir^®^-AD cream, Kobe, Japan) was applied to the shaved dorsal skin, 2 h after the 4% sodium dodecyl sulfate application (SDS, Sigma-Aldrich, St. Louis, MO, USA), twice a week, for 3 weeks in NC/Nga mice, and EVs were administered subcutaneously (SC) into the loose skin over the neck 3 times a week for 4 weeks. As a positive control, prednisolone was orally administered daily. In the seventh week, the mice were sacrificed, and skin and blood samples were collected. The experimental design is illustrated in Figure 4A.

#### 2.11.2. Clinical Observation of AD 

Clinical observation of the backs of NC/Nga mice was performed once a week for 4 weeks. To compare the skin lesions of the mice, the clinical severity of dermatitis was scored using the previously described macroscopic diagnostic criteria, which are normally used for human AD. Briefly, dermatitis severity was evaluated once per week. The development of (1) erythema/hemorrhage, (2) scarring/dryness, (3) edema, and (4) excoriation/erosion was scored as 0 (none), 1 (mild), 2 (moderate), and 3 (severe), respectively. The sum of individual scores was used as the dermatitis score. Ear thickness was measured using a caliper (Mitutoyo, 209-572M External Digital Caliper Gauge (0–20 mm, 0.01 mm); Myanmar) 4 weeks after Df-induction [25].

#### 2.11.3. Total Serum IgE Measurement

Total serum IgE levels were measured using a mouse IgE ELISA kit (ab157718; Abcam, Waltham, MA, USA), according to the manufacturer’s instructions. Serum samples were obtained by centrifugation at 12,000 rpm for 10 min and stored at −80 °C until required for the assay.

#### 2.11.4. Quantitative Real-Time PCR Analysis

After the administration of EVs for four weeks, skin lesions from the back were removed, frozen immediately in liquid nitrogen, and stored at −80 °C. The mRNA levels of the inflammatory cytokines were analyzed using RT-qPCR. RNA extraction was performed by disrupting 20 mg or less of tissue with a homogenizer, followed by RNA extraction using the AccuPrep^®^ Universal RNA Extraction Kit (Bioneer, Daejeon, Republic of Korea, Cat.K-3140) and DNase treatment using the RNase-Free-DNase Set (Qiagen, Hilden, Germany, Cat.79254). RNA quality, concentration, and purity were measured with a NanoDrop One (Thermo Fisher Scientific), and an 18S/28S peak was confirmed with a 5200 Fragment Analyzer (Agilent). RT-qPCR was performed using 48 ng RNA for the AccuPower^®^GreenStar™ RT-qPCR Master Mix (Cat.K-6403) and Exicycler TM 384 Real-Time Quantitative Thermal Block (Cat.A-2061). Primers were provided by Bioneer (Il4, 146 bp; primer no. PM00287, Gene Bank NM_021283; Il5, 110 bp, primer no. PM00178, Gene Bank NM_010558; Il13, 212 bp, primer no. PM00288, Gene Bank NM_008355; Ifng, 156 bp, primer no. PM00289, Gene Bank NM_008337; Gapdh, 146 bp, primer no. PM00118, Gene Bank NM_001289726, NM_008084).

#### 2.11.5. Histopathologic Evaluation

The tissues of the experimental mice were removed, fixed in 10% phosphate-buffered formalin, embedded in paraffin, sectioned, and stained with H&E. To measure mast cell infiltration, paraffin-embedded tissue sections were stained with toluidine blue, and the number of mast cells was counted at five random sites.

#### 2.11.6. Immunohistochemistry

Serial sections of paraffin-embedded skin were deparaffinized and incubated with an anti-TSLP (10 μg/mL; ab115700; Abcam) and CD-86(4 μg/mL; ab213044; Abcam) at 4 °C for 24 h after being blocked with normal goat serum. The slides were incubated with a biotinylated secondary antibody for 1 h, followed by avidin–biotin–peroxidase complexes (Envision kit; Dako, K5007, Glostrup, Denmark) for 2 h. Peroxidase activity was confirmed using 3,3-diaminobenzidine (Vector Labs, Newark, CA, USA) followed by a hematoxylin counterstain. The stained cells were counted in each group. The staining intensity for TSLP and CD-86 was scored on a 5-point scale as follows: 0, no positive staining; 1+, mild cytoplasmic staining; 2+, moderate-to-severe cytoplasmic staining; 3+, moderate-to-severe cytoplasmic staining with nuclear staining; 4+, severe cytoplasmic staining [25].

### 2.12. In Vivo Toxicity Studies

#### 2.12.1. Single-Dose Toxicity Study

This toxicity study began after approval of the IACUC of KCL (approval number: IA19-01714). The aim of this study was to investigate the toxicity symptoms and the approximate lethal dose. Briefly, 8-week-old, ICR mice (OrientBio, Seongnam, Republic of Korea) were randomly assigned to four groups of ten animals (five male and five female mice per group). The cASC-EVs were subcutaneously administered once each to the male and female ICR mice at doses of 7.45 × 10^8^ (low dose), 2.98 × 10^9^ (mid-dose), and 1.19 × 10^10^ (high dose) particles/20 g and compared to the vehicle control group, only treated with DPBS. During the experiment, the occurrence of dead animals, symptoms, and changes in bodyweight were noted, and the overall gross findings from the sacrificed animals were observed at the end of the experiment.

#### 2.12.2. Twenty-Eight-Day Repeat-Dose Toxicity Study

This study was conducted to evaluate the potential toxicity and the organs being targeted when the cASC-EVs were repeatedly administered to ICR female and male mice for 4 weeks. All animal experiments were approved by the IACUC of KCL (approval number: IA21-02194). Briefly, 8-week-old, ICR mice (OrientBio, Seongnam, Republic of Korea) were randomly assigned to four groups of twenty animals (ten male and ten female mice per group). The cASC-EVs were subcutaneously administered at doses of 7.45 × 10^8^ (low dose), 2.98 × 10^9^ (mid-dose), and 1.19 × 10^10^ (high dose) particles/20 g 3 times a week. Mortality, clinical observation, bodyweight change, food consumption, water intake, urinalysis, hematology, blood chemistry, necropsy findings, organ weight, and histopathological findings were evaluated during the experiment. Hematological evaluation was measured using a blood analyzer (ADVIA 2120, SIEMENS, Munich, Germany) and blood chemistry evaluation was performed using a blood biochemical analyzer (Hitachi 7180, HITACHI, Tokyo, Japan).

### 2.13. microRNA Profiling

Raw reads of small RNAs were preprocessed to eliminate adapter sequences. Adapters in the raw reads were trimmed using the Cutadapt program. The first three nucleotides of all reads were trimmed to remove extra bases that were inserted during the SMART template-switching activity process. If a sequence was matched to more than the first 5 bp of the 3’ adapter sequence, it was regarded as an adapter sequence and trimmed from the read. Trimmed reads longer than 18 bp were selected for mapping reliability. Then, the remaining reads were classified into non-adapter reads, whose adapter sequences were not sequenced. Trimmed and non-adapter reads were combined and regarded as processed reads for downstream analysis.

To minimize sequence redundancy for computational efficiency, the processed reads were clustered by a sequence. A unique cluster consists of reads with the same sequence and length. To eliminate rRNA, the reads aligned to the 45S pre-rRNA and mitochondrial rRNA from Canis lupus familiaris were excluded.

Sequence alignment and detection of known and novel miRNAs were performed using the miRDeep2 software algorithm. The rRNA-filtered reads were aligned to the mature and precursor miRNAs of Canis lupus familiaris obtained from miRBase v22.1 using the miRDeep2 quantifier module. The miRDeep2 algorithm is based on the miRNA biogenesis model; it aligns reads to potential hairpin structures to check whether their mapping context is consistent with Dicer processing and assigns scores representing the probability that hairpins are true miRNA precursors.

The reference genome of Canis lupus familiaris, released as CanFam3.1, was retrieved from RefSeq. The reference genome was indexed, and rRNA-filtered reads were mapped using Bowtie (1.1.2). Novel miRNAs were predicted from mature, star, and loop sequences, according to the RNAfold algorithm, using miRDeep2. The RNAfold function uses a nearest-neighbor Thermodynamic model to predict the minimum free-energy secondary structure of an RNA sequence.

Uniquely clustered reads were sequentially aligned to the reference genome, miRBase v22.1, and the non-coding RNA database RNAcentral release 14.0 to identify the known miRNAs and the other RNA types for classification.

For the bioinformatic analysis of miRNAs, predicted targets of miRNAs with a score of 75 or higher were selected from the miRDB database (http://miRDB.org, accessed on 30 April 2021). The gene ontology (GO) analysis of the targets was performed using DAVID 6.8 (https://david.ncifcrf.gov/home.jsp, accessed on 9 June 2023).

### 2.14. Statistical Analysis

All quantitative data are reported as means ± standard deviation. Between-group comparisons were performed using the two-tailed Student *t*-test or ANOVA, followed by Tukey’s test for normally distributed variables, or nonparametric analysis with a Mann–Whitney U-test or Kruskal–Wallis test, followed by Dunn’s multiple comparison test for nonnormally distributed variables. *p* < 0.05 was considered statistically significant.

## 3. Results

### 3.1. Characterization of Canine Adipose-Tissue-Derived Mesenchymal Stem Cells (cASC)

We isolated canine adipose-derived stem cells (cASCs) from canine adipose tissue that had been preserved for 24 h. To characterize the surface phenotype of the ASCs isolated from an aliquot preserved for 24 h, cell surface markers were examined at the third passage. Flow cytometry results showed that ASCs isolated after 24 h of preservation were positive for CD29, CD44, and CD90. In addition, the expressions of CD4, CD8, CD14, CD25, CD45, CD80, and CD184 were not observed. We confirmed the expression of CD29, CD44, and CD90 on the cASC surface using flow cytometry (Figure 1A). Our results indicated that cASCs express CD29, CD44, and CD90 specifically on the surface, similar to human ASCs. In general, the differentiation potential of stem cells was maintained during the early passages. The expression of multipotency stem cell markers in cASCs at P2 was observed using RT-PCR (Oct4, Nanog, and Sox2). The cASCs positively expressed Oct4, Nanog, and Sox2. The expression of Oct4 and SOX9 decreased with an increasing number of subpassages. However, the expression of Nanog was negatively correlated with the number of subpassages. Specific early stem cell markers include transcription factors, such as Sox2, Oct4, and Nanog (Figure 1B). As a result, it was confirmed that the expression levels of Sox2 and Oct4 decreased as the passage number increased, and the expression level of these transcription factors was higher than the control (CMT-U27), until the second passage. 

### 3.2. cASCs Have a Differentiation Phenotype for Osteogenesis, Adipogenesis, and Chondrogenesis

Differentiation into multiple mesodermal lineages is a characteristic feature of MSCs. To demonstrate multilineage differentiation potency, cASCs were cultured in osteogenic, chondrogenic, or adipogenic conditions to induce differentiation for several weeks. To determine their ability to differentiate, third passage cASCs were induced to differentiate into adipocytes, chondrocytes, and osteocytes using a modified differentiation medium for adipocytes and chondrocytes, and an osteogenic medium for osteocytes. Osteogenesis: After 21 days of culture, the cells were stained with alizarin red and analyzed for calcium mineralization. Calcium staining was evident within the differentiated cells; however, no calcium mineralization was observed in the control cells (Figure 2A). Analysis of the RT-qPCR results revealed that the mRNA expression levels of the osteogenic marker Runt-related transcription factor 2 (RUNX2) and bone morphogenetic protein 2 (BMP2) were significantly higher in differentiated cells than in non-induced control cells (Figure 2B). Chondrogenesis: After 35 days of culture, the cells were stained with Alcian blue and examined for the presence of proteoglycans. Proteoglycans were stained blue in the differentiated cells, yet positive staining was not detected in the non-induced control cells. After 35 days of pellet culturing, pellet formation was observed, whereby the pellet grew gradually. Proteoglycans were confirmed by Alcian blue (Figure 2A). The RT-qPCR results showed that the mRNA expression levels of the chondrogenic markers Sox9 and aggrecan were significantly higher in the differentiated cells (Figure 2B). Adipogenesis: After 35 days of culture, cells were stained with Oil Red O and examined for intracellular lipid accumulation. The differentiated cells showed stained lipid droplets; however, the non-induced control cells did not stain positively (Figure 2A). Based on RT-qPCR analysis, the mRNA expression levels of adipogenic markers, such as FAS and SREBP-1, were significantly higher in the differentiated cells (Figure 2B). These results suggest that cASCs have the potential to differentiate into osteocytes, chondrocytes, and adipocytes.

### 3.3. Isolation and Characterization of the cASC-Derived Extracellular Vesicles

First, we isolated canine ASC-derived extracellular vesicles (cASCs) using ExoSCRT™ technology from the cASC-conditioned media, under serum-free conditions [26]. The detailed isolation process is summarized in a schematic of the EV separation methods (Figure 3A). Characterization of EVs was performed according to the Minimal Information for Studies of Extracellular Vesicles 2018 (MISEV2018), recommended by the International Society for Extracellular Vesicles. To characterize cASC-derived EVs, we first analyzed the concentration and size distribution of the particles using an NS300 (NTA) instrument (Figure 3B,C). The results indicated that the ASC-EV mode size was approximately 103.07 +/− 6.24 nm and the average particle concentration was 9.46 × 10^9^ ± 1.72 × 10^9^ particles/mL. The average protein concentration from the three batches was 28.13/−8.80 µg/mL and the average purity of the ASC-EVs was 3.45 × 10^8^ ± 5.02 × 10^7^ particles/mL (Figure 3D,E). To further characterize the ASC-EVs, flow cytometry analysis was conducted to test the presence of tetraspanin markers known to be enriched in EVs, such as CD9, CD63, and CD81. As no antibodies reacted with canine proteins, anti-human CD9, CD63, and CD81 staining antibodies were used. The results showed that only anti-human CD81 antibodies bound to canine exosomal CD81 protein, indicating that only CD81 has cross-reactivity between humans and dogs (Figure 3F). Next, the levels of negative markers of EVs, such as calnexin, were measured (Figure 3G). The concentration of calnexin in canine ASC cell lysates was 138.43 pg/mL and the average concentration of calnexin in the three batches was 2.02 pg/mL, showing significantly lower levels of calnexin compared to the cell lysates. 

Next, we examined the anti-inflammatory effects of ASC-derived EVs. In general, ASCs have a unique immunomodulatory function [14,15], which makes them suitable for cellular therapy, such as repairing tissue damaged by chronic inflammation or autoimmune diseases. For example, they have been reported to exert beneficial effects in murine models of experimental systemic lupus erythematosus (SLE) [27] and inflammation-mediated autoimmune diseases [11,12,13]. AD is a chronic, relapsing, and highly pruritic inflammatory skin disease that significantly reduces the quality of life for patients [28,29]. As reported in our previous study on human ASC-EVs, systemic administration of ASC-EVs ameliorated AD-like symptoms through the regulation of inflammatory responses and the expression of inflammatory cytokines [26]. With the same flow, canine ASC-derived EVs have similar effects on the regulation of the inflammatory response and expression of inflammatory cytokines. 

### 3.4. Effect of EVs on Improvement of Biostir-Induced Atopic Skin Lesions in Mice

To investigate the effect of EVs on Biostir-induced AD, all animals were clinically observed on a weekly schedule. Df-stimulated back skin from NC/Nga mice developed AD-like skin lesions. We found that the SC administration of EVs significantly decreased AD symptoms. Compared to the negative control group, the skin condition of the EV-treated groups was improved, and the thickness of the ear increased by Biostir decreased as well (Figure 4B,D). In addition, the clinical evaluation scores of the EV treatment groups were significantly reduced (Figure 4C). The concentration of total plasma IgE tends to be high in allergic patients, such as those with AD, and is known to increase with disease onset and exacerbation [26]. We measured the total plasma IgE levels in mice with AD induced by Biostir using an ELISA kit. As a result, an increased level of IgE was also used as a diagnostic indicator of AD. Df significantly upregulated total plasma IgE production induced by Df stimulation. As shown in Figure 4E, the EV treatment group and the positive control group showed significantly downregulated IgE levels in NC/Nga mice (Figure 4E). The mRNA levels of inflammatory cytokines were analyzed using RT-qPCR. The systemic administration of EVs dose-dependently reduced the upregulated mRNA levels of IL-4 and IFN-ɣ-in the skin lesions compared to the vehicle control; the reduction was comparable to that with prednisolone treatment (Figure 4F,G). It was shown that the dorsal skin of NC/Nga in the AD group (G2) had distinct hyperplasia and hyperkeratosis in the thickened epidermis and infiltrated inflammatory cells, which were attenuated by EVs and prednisolone treatment, reducing the thickness of the epidermis in a dose-dependent manner (Figure 5). 

**Figure 4 animals-13-02215-f004:**
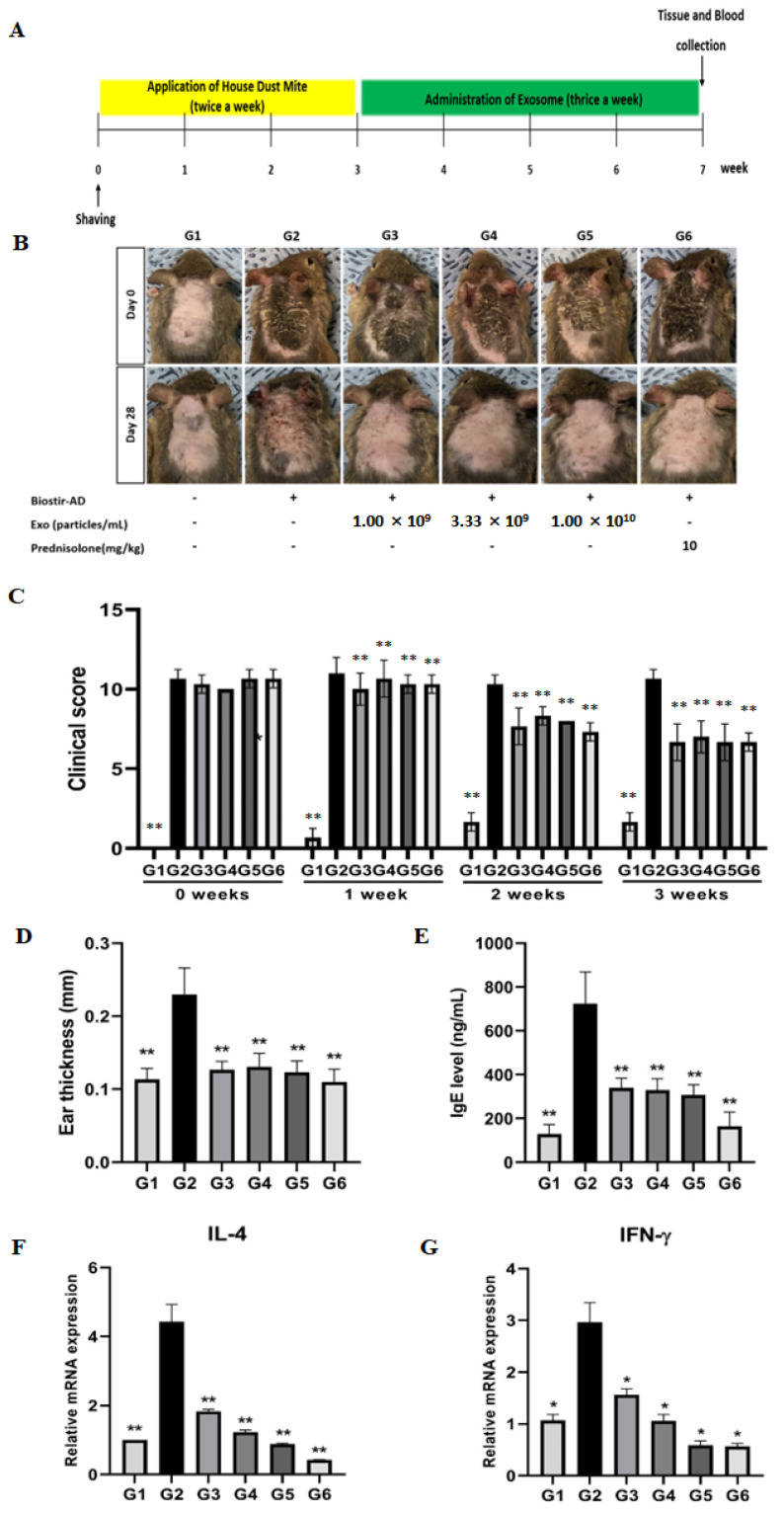
cASC-EVs alleviate dermatitis in Biostir-induced atopic dermatitis-like skin lesions. (**A**) Schedule of the in vivo animal experiments. (**B**) Photographs of the dorsal lesion area of mice on days 0 and 28. (**C**) Clinical evaluation using the score index of AD, each week for four weeks. (**D**) Ear thickness in mice with Biostir-induced AD-like skin lesions. (**E**) IgE levels in the total plasma of Biostir-induced AD mouse model. (**F**,**G**) IL-4 and IFN-gamma mRNA levels, measured by RT-qPCR (*n* = 3). Data are mean ± SD. ** *p* < 0.01 vs. group 2, * *p* < 0.05 vs. group 2. G1 = Negative control group, G2 = Biostir-induced AD group, G3 = Biostir-induced AD + cASC-EV 1.00 × 10^9^ treated group, G4 = Biostir-induced AD + cASC-EV 3.33 × 10^9^ treated group, G5 = Biostir-induced AD + cASC-EV 1.00 × 10^10^ treated group, G6 = Biostir-induced AD + prednisolone 10 mg/kg treated group.

### 3.5. cASC-EVs Ameliorate the Signs of AD In Vivo

To investigate whether EVs ameliorate AD symptoms in vivo, we evaluated the histological effects of EVs in a murine model. We found that the administration of EVs significantly decreased epidermal thickness, and the manifestations of mast cells including, TSLP, and CD86, as shown in the histology (Figure 5A). Consistently, the number of mast cells decreased in the EV-treated mice (Figure 5B). We also found that the intensity of TSLP and CD86 in the skin lesions was significantly decreased by the administration of EVs (Figure 5C,D). Interestingly, it has been reported that T cells and inflammatory dendritic epidermal cells, which are not found in normal mice but are abundant in AD mice, express both TSLP and CD86 on their surfaces. These results indicated that cASC-derived EVs significantly decreased AD symptoms compared to negative control group.

### 3.6. In Vivo Toxicity Studies

#### 3.6.1. Single-Dose Toxicity Study

No dead animals or notable symptoms were observed during the experiment. However, the analysis of weight measurements indicated weight loss in some animals, although statistical tests showed no statistically significant difference between groups, and it was judged to be a transient change between individuals that had no toxicological significance. As a result of the necropsy at the end of the experiment, no gross findings were observed in any of the tested animals.

Based on the above results, no dead animals were observed when the cASC-EVs were administered once to an ICR mouse, and thus, the approximate lethal dose of cASC-EVs under this test condition is judged to be 1.19 × 10^10^ particles/20 g or more.

#### 3.6.2. Twenty-Eight-Day Repeat-Dose Toxicity Study

During the experimental period, no animal deaths were observed in any treated group. The cASC-EV-related changes were not detected in any treated groups when analyzing clinical observations, bodyweight, feed consumption, water intake, and urinalysis parameters.

There were no cASC-EV-related changes in the hematology parameters. Some differences showed statistically significant increases in EO (eosinophil), EOP (percent of eosinophil), and LUP (percent of the large unstained cell) levels in the male group compared to the control group, and significantly decreased RBC (red blood cell), Hb (hemoglobin), and HCT (hematocrit) levels in the male group. Moreover, EO, LUC (large unstained cell), and MPV (mean platelet volume) in the female group were statistically significantly increased compared to the control group. However, there was no toxicological significance because there was no dose-response correlation or minimal change within the range of biological fluctuations with a slight difference in degree (Appendix A).

There were no cASC-EV-related changes in the blood chemistry parameters. The PRO (total protein) level in the males decreased statistically significantly compared to the control group, and the Cl (chloride) level in the males increased statistically significantly. The Na (sodium) level in the females decreased statistically significantly. However, these differences were attributed to biological variation because the degree was minor and the dose-response correlation was not clear; therefore, there was no toxicological significance (Appendix A).

In organ weight change, the relative ovary (right) weight in the female animals treated with 2.98 × 10^9^ particles/20 g was statistically significantly lower than in the control group, but the dose-response correlation was not clear and notable findings were not observed in the histopathological evaluation.

In the scheduled necropsy, no gross findings were noted in any of the terminal animals and no cASC-EV-related microscopic findings were noted. The microscopic findings observed were considered incidental to the nature commonly observed in mice and thus were considered unrelated to the administering of the cASC-EVs.

### 3.7. MicroRNA Profiling of cASC-Derived EVs

We analyzed the expression of 798 miRNAs in two batches of cASC-EVs. Heatmaps of the expressed miRNAs were applied between the two batches to identify statistically significant differentially expressed miRNAs (Figure 6A). Twenty highly expressed miRNAs (|FC| > 2 and cutoff *p* ≤ 0.05) were analyzed. Using heat map analysis, it was possible to observe highly expressed miRNAs in the gene expression profile (blue (down) and red (up)) between the two batches (Figure 6B). In the profiling data, the top 20 miRNAs accounted for 87% of the total miRNAs and the top 7 miRNAs (let-7a, let-7b, miR-21, let-7f, miR-125b, miR-24, and miR-29a) consisted of 66% of the total miRNAs. Interestingly, recent studies have confirmed a key role for miR-21 and let-7b in the resolution of inflammation and in negatively regulating the proinflammatory response induced by many of the same stimuli that trigger miR-21 and let7b induction itself. Further, miR-21 has emerged as a key mediator of anti-inflammatory responses in macrophages.

The results of the GO analysis performed on the targets of cASC-EV miRNAs revealed that the biological process of the targets is primarily associated with “intrinsic apoptotic signaling pathway in response to DNA damage” processes (Figure 6C), the molecular function is “protein serine/threonine kinase activity” (Figure 6D), and the pathway most strongly associated is the “JAK-STAT signaling pathway” (Figure 6E).

## 4. Discussion

Human MSCs are known to regulate inflammatory relief and immune activation through the secretion and interaction of EVs, growth factors, and various cytokines [30,31]. In particular, MSC-EVs are known to contain abundant substances such as proteins, lipids, and miRNAs, and are attracting attention as a leader in cell-free therapy as an alternative to cell therapy [32]. Recent studies have reported that it is related to various immunological diseases, such as allergic reactions, asthma, and AD [33]. Additionally, various studies have shown that MSC-EVs play a major role in immunosuppression. In the field of cell therapy, the development of treatments using MSCs is an important topic, yet issues related to immunogenicity and side effects are emerging as the biggest problems in cell therapy [34]. The development of treatments using MSC-EVs has been suggested as a good alternative to reduce the side effects [35]. Therefore, many studies have been conducted to alleviate symptoms of AD, such as itchiness, skin barrier defects, and inflammation through MSC-EVs [36].

However, in the case of canine AD, few studies have investigated treatments using canine MSCs. Therefore, there are few treatments for AD or other inflammatory diseases using canine-MSCs [24]. We are interested in the treatment of inflammation using cASC-EVs, particularly in the treatment of atopy. Here, we separated canine MSCs from adipose tissue, verified the cells, and isolated EVs from the cells. We previously identified the anti-inflammatory effect of human MSC-EVs by investigating the therapeutic effect of AD from human MSC-EVs [26]. Accordingly, we hypothesized that confirming the anti-inflammatory effect of canine MSC-EVs could lead to the same concept. Previously, there were only a few studies using characteristic canine AD models other than DNBC-treated models, and it was expected that using cASC-EVs would have a promising effect on the treatment of canine AD [24].

In the Biostir-induced AD model, we confirmed both a reduction in inflammation-related factors and the reduction of AD-related levels in a dose-dependent manner following treatment with cASC-EVs. This result indicates that the inflammatory environment of AD mice is regulated by the activation of T cells and mast cells, thereby reducing the inflammatory response in the Biostir-induced AD model. In addition, the reduction of serum inflammation-related cytokines, such as IL-4 and IFN-gamma, demonstrated the effectiveness of cASC-EVs in the treatment of AD. In general, IL-4 and IFN-gamma strongly inhibit the expression of barrier-related molecules that play an important role in maintaining the structural integrity and function of the stratum corneum of the skin and allow the stratum corneum to be maintained [37]. Through this, it was confirmed that the reduction in the same inflammation-related factors occurred in both human ASC-EVs and in dogs.

In addition, the increase in serum IgE levels in the Biostir-induced AD model was remarkably reduced by treatment with cASC-EVs. This result indicated that cASC-EVs inhibited skin inflammation in the Biostir-induced AD model through T-cell activation and mast cells. Thus, it was possible to confirm the regulation of immune cell activation by cASC-EVs. Similarly, TSLP and CD86 [38], which are indicators of inflammatory diseases, including in AD models, were decreased in a concentration-dependent manner by cASC-EVs, confirming that cASC-EVs exhibit anti-inflammatory effects.

From the single-dose and 28-day repeat-dose toxicity studies, the potential toxicity and target organs were not observed when cASC-EVs were subcutaneously administered to ICR mice under test conditions. Therefore, the NOAEL (no observed adverse effect level) of cASC-EVs is considered to be 1.19 × 10^10^ particles/20 g (the highest dose).

Although EVs have been studied for many years, the biological roles of EV-miRNAs have only recently been investigated [39]. Our data showed that 798 miRNAs were identified and profiled using thermal map analysis. We analyzed the top 20 most highly expressed miRNAs in cASC-EVs. Among the various miRNAs, let-7a, let-7b, and miR-21 were highly expressed in cASC-EVs. These miRNAs, which are abundant in cASC-EVs, are known to exert anti-inflammatory effects in recipient cells through miRNA transfer [40,41,42]. Since let-7b and miR-21 have been shown to regulate inflammation in various contexts, our findings provide new insights into how and where they function, allowing us to better understand how they regulate inflammation-related processes. Furthermore, our study adds to the promising body of evidence which highlights that miRNA transfer within EVs is part of an intercellular communication network that coordinates complex immune responses [43,44].

## 5. Conclusions

We successfully isolated canine adipose stem cells and EVs in this study. It was shown that cASC-EVs have a beneficial effect in an AD model and also demonstrated that cASC-EVs provide no safety concerns in single- or repeat-dose toxicity studies, and we identified the role of EV miRNAs in improving anti-inflammatory function using NGS. Therefore, these findings can serve as a basis for the treatment of AD and suggest that cASC-EVs may provide a novel therapeutic approach to treating canine AD.

## Figures and Tables

**Figure 1 animals-13-02215-f001:**
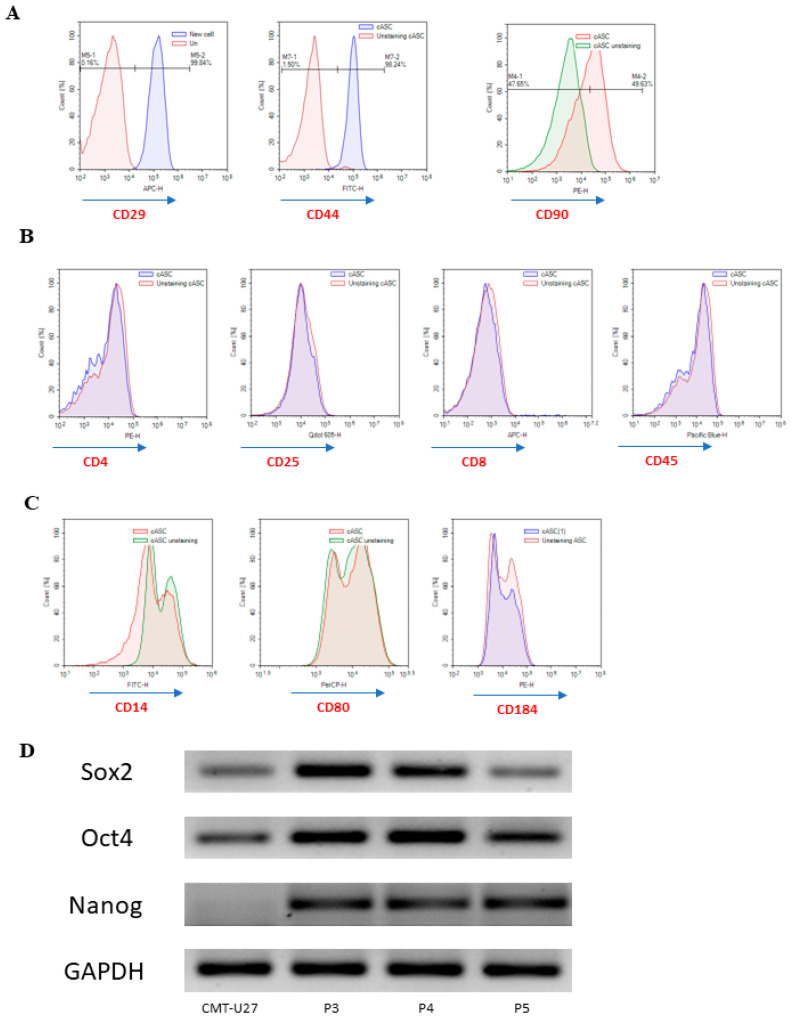
Characterization of cASCs. (**A**–**C**) Expression of cell surface markers of cASCs isolated from adipose tissue preserved for 24 h, determined by flow cytometry. (**D**) Expression of multipotency markers (Sox2, Oct4, and Nanog) by RT-PCR in cASCs during continuous passages. GAPDH was used as an internal control. The cASCs were positive for all markers. CMT-U27 (canine mammary cancer cell line) is used for comparison.

**Figure 2 animals-13-02215-f002:**
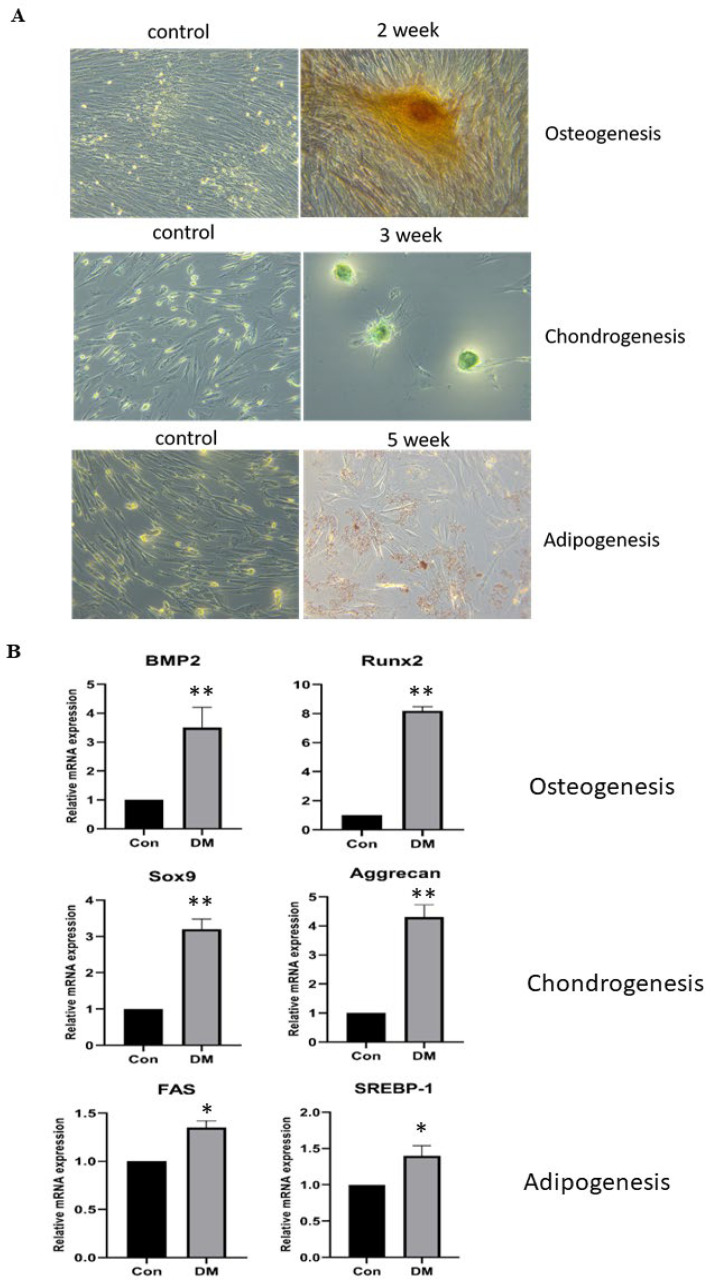
Differentiation of canine adipose-derived mesenchymal stem cells (cASCs) into osteocyte, chondrocyte, and adipocyte lineages. (**A**) Images of alizarin red staining for osteogenic lineage. Images of Alcian blue staining for chondrogenic lineage. Images of Oil Red O staining for adipogenic lineage. (**B**) Reverse-transcript quantitative PCR (RT-qPCR) data on gene expression relating to osteogenic, chondrogenic, and adipogenic factors (BNP2 and RUNX2 for osteogenic, Sox9 and aggrecan for chondrogenic, alongside FAS and SREBP1 for adipogenesis, respectively; *n* = 3 for each lineage). Data are mean ± SD. * *p* < 0.05 vs. Control group, ** *p* < 0.001 vs. Control group.

**Figure 3 animals-13-02215-f003:**
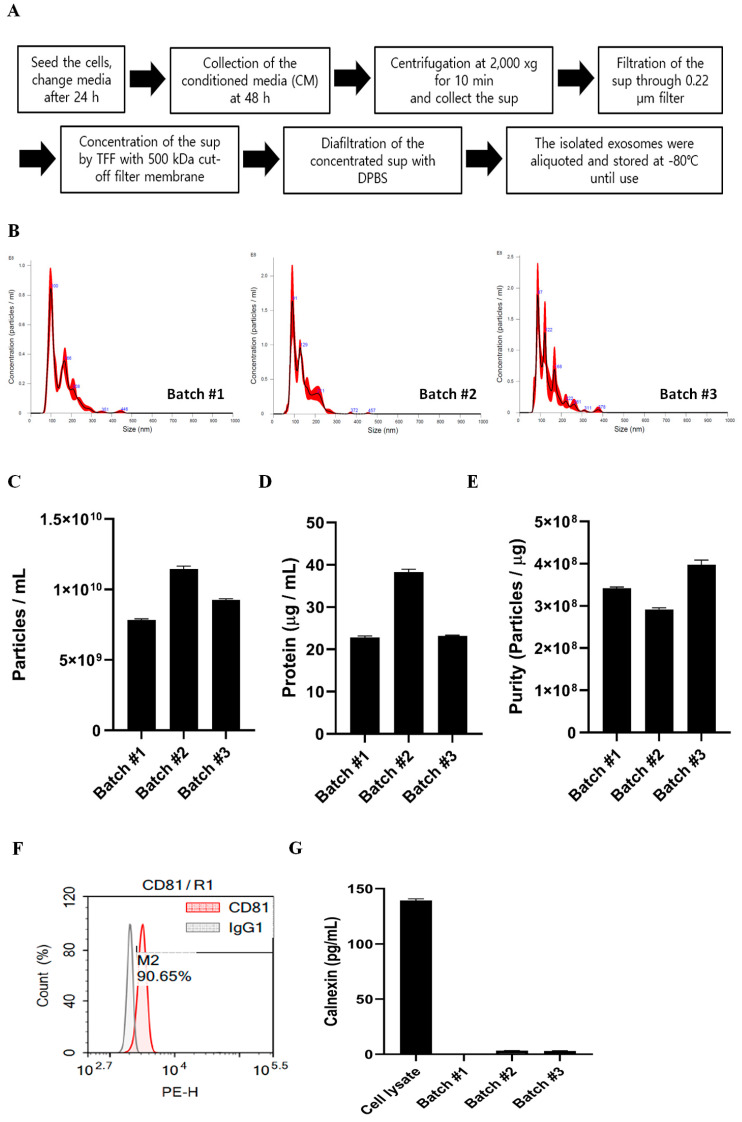
Isolation and characterization of canine ASC-EVs. (**A**) The schematic summary of the EV separation methods. (**B**) Histogram of particle concentration and size distribution of canine ASC-EVs measured by nanoparticle tracking analysis (NTA) (*n* = 3). The red part of the histogram represents the range of deviation. (**C**–**E**) Particle concentration, protein concentration, and purity of canine ASC-EVs (*n* = 3). (**F**) Histograms showing canine ASC-EVs stained with anti-human CD81 antibody, which are shown in comparison to an IgG1 isotype-stained negative control. (**G**) Calnexin and concentration of canine ASC-EVs in each batch (*n* = 3).

**Figure 5 animals-13-02215-f005:**
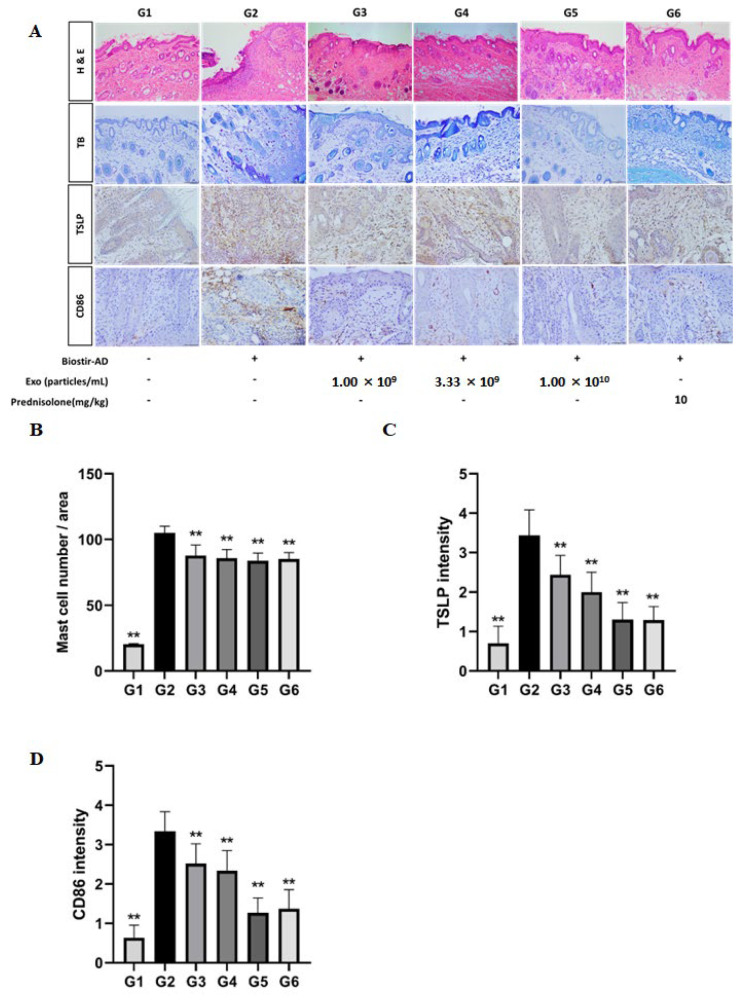
Inhibitory effects of EVs on AD-like lesions in Biostir-AD induced atopic dermatitis-like phenotypes in a murine model. (**A**) Representative H&E (hematoxylin & eosin) and TB (toluidine blue) staining images of dorsal skin in normal mice and mice with Biostir-induced AD topically treated with EVs in a dose-dependent manner (1.00 × 10^9^, 3.33 × 10^9^, and 1.00 × 10^10^ particles/mL) and prednisolone (10 mg/kg; positive control). Representative immunohistochemical staining images revealed infiltration of TSLP and CD86 in dermatitis. Positively stained cells are shown in brown. (**B**) Graph indicating the number of infiltrated mast cells in the tissue, determined by toluidine blue staining. (**C**) Graph indicating the intensity of immune-positive cells of TSLP expression in dermatitis. (**D**) Graph indicating the intensity of immune-positive cells of CD86 expression in dermatitis. Data are mean ± SD. ** *p* < 0.01 vs. group 2. G1 = Negative control group, G2 = Biostir-induced ADgroup, G3 = Biostir-induced AD + cASC-EV 1.00 × 10^9^ treated group, G4 = Biostir-induced AD + cASC-EV 3.33 × 10^9^ treated group, G5 = Biostir-induced AD + cASC-EV 1.00 × 10^10^ treated group, G6 = Biostir-induced AD + prednisolone 10 mg/kg treated group.

**Figure 6 animals-13-02215-f006:**
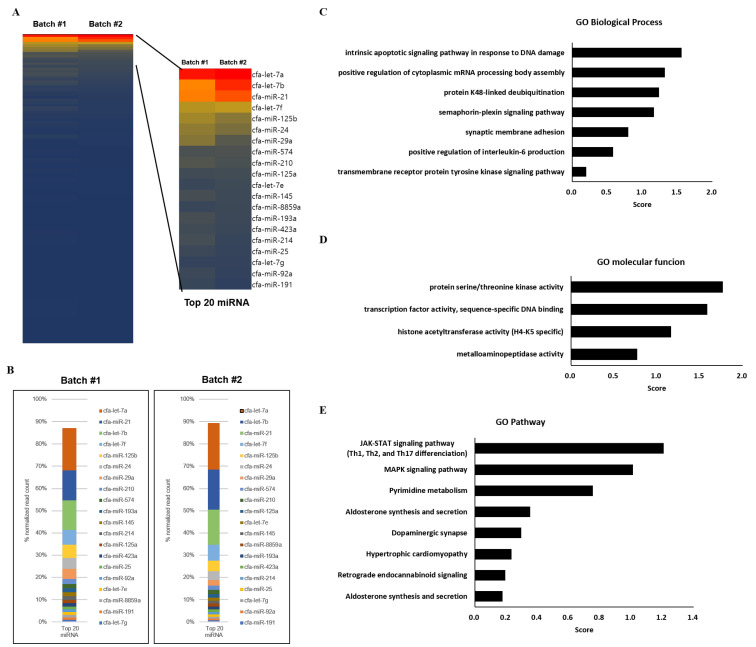
miRNA expression profiles in different batches of cASC-derived EVs. (**A**) The heatmap describes the total miRNA profiles in EVs. (**B**) The ratio of different miRNAs in the top 20 miRNAs in EVs. (**C**–**E**) Gene Ontology (GO) analysis of the targets of cASC-EV miRNA. Enrichment of GO biological process, molecular function and pathway performed using DAVID Bioinformatics resources 6.8.

## Data Availability

All the data of the study can be made available upon request.

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
