# Peer review of "Canine Mesenchymal-Stem-Cell-Derived Extracellular Vesicles Attenuate Atopic Dermatitis"

_animals, 2023, doi:10.3390/ani13132215_

Round 1

Reviewer 1 Report

This study examined the efficacy of adipose-derived mesenchymal stem cell exosomes (cASC-EVs) in reducing inflammation in canine atopic dermatitis (AD). The researchers performed in vivo toxicity studies to assess potential side effect of cASC-EVs. Results showed that cASC-EVs ameliorated AD -like dermatitis and reduced inflammatory cytokines and chemokines , with no toxicity detected. In addition, miRNA arrays of cASC-EVs were analyzed, suggesting that anti-inflammatory miRNAs were responsible for the observed improvements. Overall, cASC-EVs offer a promising cell-free, non-toxic therapeutic option for the treatment of dogs AD.

The work was performed at a good methodological level. However, most of the methods do not include enough information to allow an independent researcher to replicate the data or perform a critical analysis. The section needs improvement.

It is necessary to include the breed of the dogs as well as the age of the adipose tissue donors because age is known to affect the efficacy of cell therapy (PMID: 34063923). How many individual dog donors were used to obtain MSC cultures.

Please indicate at which passage MSC immunophenotyping was performed

Please summarize the three methodological sections "2.3. Reverse transcription polymerase chain reaction (RT-PCR)" and "2.7. Real-time quantitative polymerase chain reaction (RT -qPCR)". It is necessary to provide a list of primers used in the analysis of MSCs.

The passage at which the conditioned culture medium was taken and the number of days the medium was conditioned must be provided. How was the culture medium used to prepare the EV?

What method was used to measure the protein concentration in the EVs?

Describe in detail the procedure for subcutaneous administration of the EVs.

How many animals were included in the acute and chronic toxicity study?

The section describing the research results needs clarification.

Provide a legend for the graphs in Figure 4C-G.

Did the data in Figure 4С change in a statistically significant way?

Figure 5A needs to be enlarged because it is impossible to distinguish the histological structures.

It is necessary to indicate the statistical criteria used when presenting data on figures.

The authors write that extracellular vesicle therapy caused a decrease in epidermal thickness, whether morphometric analysis was performed?

The following statement is not consistent with the data presented, "These results indicated that cASC-derived EVs significantly decreased AD symptoms in a dose-dependent manner" ( line 475). The authors need to demonstrate statistically significant differences between groups treated with different concentrations of extracellular vesicles.

On what basis do the authors make the following assumption "Based on the above results, no dead animals were observed when the cASC-EVs were administered once to an ICR mouse, and thus, the approximate lethal dose of cASC-EVs under this test condition is judged to be 1.19E+10 particles/20 g or more" which does not follow from their experimental study?

The authors are advised to move the supplemental section Table 1-1 and Table 2-1 as they clutter the main body of the article. It is necessary to describe the methodology used to evaluate the blood hematological and biochemical parameters .

Bioinformatic analysis of molecular functions, cellular components, biological processes, and pathways will add value to the miRNA  data.

Author Response

Please see the attachment  including the revised manuscript and cover-letter with response to the reviewer`s comments.

Reviewer 2 Report

1) Authors in the methods section please detail how many passages occurred during the expansion of the adipose MSCs.

2) Much debate occurs in the literature on the CD markers for MSCs. Which society or source did you decide on with respect to the "stemness" markers?

3) Please detail what the final passage of the conditioned media for treatment was. For instance on page 12 in the flow diagram you state collect the CM after 48 hours. Was this is same time frame (48 hours) as the analyzed CM to the treatment media. The analyzed CM should be the same (same passage  expansion) as the CM used to treat AD, unless justification is provided. In all I am confused on which passage of CM was analyzed and then which was used to treat the AD in vivo.

Over all this is a very impressive study with verification of the MSCs, potency and treatment data.

Author Response

(The authors gave the same response as above.)

Round 2

Reviewer 1 Report

Dear Authors,

I am pleased to inform you that after reviewing the revised version of your manuscript, I have found the changes and additional data to be well-addressed and have incorporated them. The authors have provided a satisfactory response to all previous concerns and have significantly improved the scientific rigor and quality of the manuscript.

As a result, I believe that the updated manuscript has significantly improved the study's findings and overall presentation of the study, making it suitable for publication in Animals.